

# Detecting racism and xenophobia using deep learning models on Twitter data: CNN, LSTM and BERT

José Alberto Benítez-Andrades[1], Álvaro González-Jiménez[2], Álvaro López-Brea[2], Jose Aveleira-Mata[3], José-Manuel Alija-Pérez[3] and María Teresa García-Ordás[3]

[1] SALBIS Research Group, Department of Electric, Systems and Automatics Engineering, Universidad de León, León, León, Spain
[2] Department of Electric, Systems and Automatics Engineering, Universidad de León, Leon, León, Spain
[3] SECOMUCI Research Group, Escuela de Ingenierías Industrial e Informática, Universidad de León, León, León, Spain

## ABSTRACT

With the growth that social networks have experienced in recent years, it is entirely impossible to moderate content manually. Thanks to the different existing techniques in natural language processing, it is possible to generate predictive models that automatically classify texts into different categories. However, a weakness has been detected concerning the language used to train such models. This work aimed to develop a predictive model based on BERT, capable of detecting racist and xenophobic messages in tweets written in Spanish. A comparison was made with different Deep Learning models. A total of five predictive models were developed, two based on BERT and three using other deep learning techniques, CNN, LSTM and a model combining CNN + LSTM techniques. After exhaustively analyzing the results obtained by the different models, it was found that the one that got the best metrics was BETO, a BERT-based model trained only with texts written in Spanish. The results of our study show that the BETO model achieves a precision of 85.22% compared to the 82.00% precision of the mBERT model. The rest of the models obtained between 79.34% and 80.48% precision. On this basis, it has been possible to justify the vital importance of developing native transfer learning models for solving Natural Language Processing (NLP) problems in Spanish. Our main contribution is the achievement of promising results in the field of racism and hate speech in Spanish by applying different deep learning techniques.

## INTRODUCTION

In recent years, the use of social networks such as Twitter, Facebook or Instagram, as well as other community forums, is generating hateful conversations among users (*Del Vigna et al., 2017*; *Watanabe, Bouazizi & Ohtsuki, 2018*; *Zhang & Luo, 2019*). The fact that people can comment anonymously is one of the factors that leads to the spread of hate speech (*Barlett, 2015*).

Corresponding author
José Alberto Benítez-Andrades, jbena@unileon.es

Within hate crimes, some countries are particularly interested in studying racist and xenophobic crimes (_Sayan, 2019_; _Rodríguez Maeso, 2018_). For example, in the UK, hate speech towards different Muslim and other immigrant communities has increased (_BBC News, 2017_; _Travis, 2017_). A link was found between the rise of racism and the exit from the EU or the Manchester bombings. In other countries, such as Spain, the Congress of Deputies has recently approved the "Non-legislative motion on preventing the spread of hate speech in the digital space" (_Congreso de los Diputados of Spanish Government, 2020_). In short, the fight against racism and xenophobia are issues of international concern.

Some studies are currently working on the classification of texts for the detection of racism or xenophobia (_Kumari et al., 2021_, _Al-Hassan & Al-Dossari, 2019_; _Alotaibi & Abul Hasanat, 2020_, _Plaza-Del-Arco et al., 2020_; _Chaudhry, 2015_; _Konstantinidis, Papadopoulos & Kompatsiaris, 2017_; _Sazzed, 2021_). To do this, they have used different techniques in the field of artificial intelligence and, more specifically, within natural language processing. One of the most widely used techniques for this purpose is sentiment analysis on different datasets obtained from some social networks, Twitter being one of the most widely used (_Saha, Senapati & Mahajan, 2020_; _Garcia & Berton, 2021_).

In order to classify texts using machine learning techniques and, more specifically, deep learning, techniques are often used such as Convolutional Neural Network (CNN), Recurrent Neural Network (RNN) and Hierarchical Attention Network (HAN). Within these techniques, Long Short-Term and, more specifically, Bi-directional Long Short-Term (BiLSTM) are the most responsive. Analysing the latest developments in this regard are the BERT-based approaches.

However, one of the barriers encountered in detecting racist and xenophobic tweets is language. There are deep learning models that detect racist or xenophobic texts in English. Nevertheless, so far, no pretrained model has been found with texts in Spanish and there are studies showing that classification models trained with native language texts give better results (_Gutiérrez-Fandiño et al., 2021_; _Pomares-Quimbaya, López-Úbeda & Schulz, 2021_; _Kamal, Kumar & Vaidhya, 2021_; _Sharma, Kabra & Jain, 2022_; _Velankar et al., 2021_). Therefore, the main objectives of this research are those shown below:

- Obtain a set of texts in Spanish related to racism and xenophobia. Preprocess the data set and label said texts in a binary category, giving a value of 1 if the message is xenophobic or racist, and 0 when it is not.
- Apply different deep learning models and determine which of them is the one with the highest accuracy and precision.

In this paper, we propose a dataset related to racism and xenophobia in the Spanish language and we propose different models based on deep learning techniques to detect racist or xenophobic texts.

The paper is organized as follows. Related work is presented in "Related Work". The methodology of the different proposed techniques is detailed in "Methodology". In "Experiments and Results", the setup of different deep learning techniques and the results, showing a comparative between the different techniques are explained. Finally, results have

been discussed in "Critical Evaluation and Comparison of the Results" and we conclude in "Conclusions".

# RELATED WORK

Within the state of the art, two different branches can be analysed, the Spanish language datasets publicly available on the web and the natural language processing techniques most commonly used in text classification.

## Existing datasets

While it is possible to find a large number of datasets in English that are properly labelled for the detection of racism, and especially hate speech, it is extremely difficult to find datasets in Spanish with these same characteristics.

A total of two Spanish datasets related to hate speech has been found, and it is completely impossible to find one related solely and exclusively to racism/xenophobia. Nevertheless, both datasets have been explored: HaterNet (*Pereira-Kohatsu et al., 2019*) and HatEval (*Basile et al., 2019*).

Other datasets were found that also contained texts in Spanish, but were not exclusively in Spanish, as in the case of the PHARM project, where the dataset is composed of texts in Spanish, Italian and Greek (*Vrysis et al., 2021*).

### HaterNet

The first of the two datasets has been obtained thanks to HaterNet (*Pereira-Kohatsu et al., 2019*), an intelligent system used by the National Office for the Fight against Hate Crimes, belonging to the Spanish Ministry of Interior (*Ministerio del Interior, 2019*).

This dataset has a total of 6,000 tagged tweets indicating the presence or not of hate speech. Among the hate speech, there are, unsurprisingly, some tweets in which racism can be seen. However, they represent a rather low percentage of the total number of labelled tweets, which in practice makes the dataset invalid for training the model proposed in this project.

Not only are there tweets on topics that differ completely from racism, but the labelling criteria are also somewhat confusing. For these reasons, this dataset has been completely discarded for use in the training and subsequent validation of the predictive model.

### HatEval

The second dataset has been obtained thanks to HatEval (*Basile et al., 2019*), a competition organised by SemEval (International Workshop on Semantic Evaluation), whose main objective was to develop a predictive model for the detection of anti-female and anti-immigration hate speech on Twitter. In order to access this dataset, it is necessary to request permission from one of its creators as shown in the repository https://github.com/msang/hateval.

The dataset consists of two different sets of tweets. The first one contains tweets written in English, which makes it completely unusable for the purpose of this project. The second one, in turn, contains 5,000 tweets written in Spanish in the training set and 1,600 in the test set.

Of these 5,000 tweets, a total of 1,971 are related to racism, which makes the set, at least initially, valid.

As regards the structure of the dataset, for each tuple, a text followed by three binary variables can be found. The first, called HS, denotes whether the tweet expresses hate speech or not. The second, known as TR, indicates whether the victim is a single person or a group of people. Finally, the variable called AG indicates whether the tweet is expressed in an aggressive way or not.

After looking closely at the dataset, it was determined that 987 (50.07%) tweets out of 1,971 were mislabelled. There are tweets from some people denouncing racism that are labelled as racist. The same goes for some comments that are clearly ironic. On other occasions, tweets that have nothing to do with racism (in South America the word "sudaca" is used to refer to the subcontinent itself) or that are simply not racist are labelled as hate crimes.

Because of all these labelling failures, it has been determined that this dataset is also not valid for the purpose of this project.

Therefore, after having thoroughly analysed both datasets, it has been concluded that it is strictly necessary to build a suitable dataset. The whole process of building this dataset can be found in "Methodology".

## Text classification

The techniques that can be used to create models of automatic classification of a text are very varied. However, it is possible to group them into three main types of techniques: conventional machine learning, deep learning and transfer learning.

### Conventional machine learning

Among the various conventional machine learning techniques used in text classification, and especially in the detection of hate speech on the Internet, support vector machines (SVM) and logistic regression stand out.

SVMs (*Cortes & Vapnik, 1995*) are extremely accurate and effective in classifying texts (*Ahmad et al., 2018*; *Hasan, Maliha & Arifuzzaman, 2019*). One of their main advantages is that, unlike most other techniques, they perform particularly well on small training sets. Commonly, when using this technique, text features are usually obtained by applying TF-IDF or word embeddings, or another technique with a similar functionality such as Part of speech (POS).

Logistic regression (*Lakshmi, Divya & Valarmathi, 2018*; *Br Ginting, Irawan & Setianingsih, 2019*) is a type of regression analysis used to predict the outcome of a categorical variable given a set of independent variables. Like SVMs, it is also necessary to extract text features with one of the aforementiond techniques before training begins. Unlike SVMs, they perform significantly worse on small training sets.

### Deep learning

Deep learning is a specific branch of machine learning whose main difference is that, unlike traditional machine learning algorithms, it is capable of continuing to learn as it receives more and more data, without stagnating.

Among the deep learning techniques most commonly used in text classification, convolutional neural networks (CNNs) and recurrent neural networks (RNNs) stand out.

Convolutional neural networks (*Nedjah, Santos & de Macedo Mourelle, 2019*; *Roy et al., 2020*) are a type of neural network that, combined with supervised learning, process layers by mimicking the human eye, allowing them to differentiate different features in the received inputs. Although they were specifically designed for computer vision, they have been shown to perform excellently on textual classification problems as well. When extracting text features, a stage prior to learning the network, it is common to use word embeddings.

Recurrent neural networks (*Paetzold, Zampieri & Malmasi, 2019*) are a type of neural network in which the connections between nodes form a directed graph along a temporal sequence. Among the different variants of this type of network, the Long Short-Term Memory Network (LSTM) (*Talita & Wiguna, 2019*; *Bisht et al., 2020*; *Zhao et al., 2020*; *Zhang et al., 2021*), which were specifically designed to avoid the problem of long-term dependency. As with CNNs, a correct extraction of text features prior to the learning period of the network, which is carried out by means of word embeddings, is of vital importance.

### Transfer learning

Transfer learning is a machine learning technique in which the knowledge gained from carrying out a certain task is stored and then applied to a related problem. It is especially used for building models where a small amount of data is available for training and evaluation.

Within NLP tasks, the first attempted use of transfer learning was the creation of embeddings from large datasets such as Wikipedia. Although this was a major breakthrough in NLP, especially when training on very small datasets, there were still problems in differentiating the context in which words are written.

To solve this problem, a number of context-based pre-trained models such as Embedding from Language Models (ELMO) (*Peters et al., 2018*) and Bidirectional Encoder Representations from Transformers (BERT) (*Devlin et al., 2019*) have emerged, and a model very similar to the latter will be used in this project.

## Detecting racism and hate speeches in Spanish

As regards the detection of racism, to date there is no model in Spanish developed for this specific purpose. However, research has been carried out on hate speech in Spanish.

The most important study on hate speech in Spanish is (*Plaza-Del-Arco et al., 2020*). It compares the performance of various NLP models in classifying tweets in the HatEval dataset. These models include SVM, linear regression (LR), Naive Bayes (NB), decision trees (DT), LSTM and a lexicon-based classifier. Of these, the best performing was an Ensemble Voting Classifier, which combines the output of several classifiers when making a prediction.

The second study (*del-Arco et al., 2021*), by the same authors, compares the performance of other NLP models when classifying tweets from the HaterNet and HatEval

datasets. These models include deep learning models such as LSTM, Bidirectional Long Short-Term Memory Networks (BiLSTM) and CNN, as well as transfer learning models such as mBert, XLM and BETO (*Peters et al., 2018*), all of which are based on BERT. The best performer on almost all metrics was BETO.

Finally, the study (*Pereira-Kohatsu et al., 2019*) presents HaterNet, an intelligent system that identifies and monitors the evolution of hate speech on Twitter. In this study, in addition to building the aforementioned HaterNet dataset, a series of comparisons are also made between different text classification models. These models include LDA, QDA, Random Forest, Ridge Logistic Regression, SVM and an LSTM combined with an MLP, the latter being the best performing.

After having analysed the different existing models, and having verified that there are no models specifically developed to detect racism in texts written in Spanish, and that the hate speech detection models have all been trained on the same two datasets, which have numerous mislabelled tweets, the development of new models for this purpose is more than justified.

## Summary of the literature review

A summary of the literature review is shown in the Table 1, specifying the important results found in each study analysed, as well as the weaknesses found in relation to the objectives presented in our study.

# METHODOLOGY

This section explains the methodology used in this research. In Fig. 1 it is possible to see the complete workflow of the work carried out.

All the data used in the experiments, as well as the models generated and the source code are available in a GitHub repository at this link: https://doi.org/10.5281/zenodo.5188098.

## Dataset

Having found that neither of the two public datasets in Spanish labelling the presence or absence of hate speech was sufficiently valid for training the proposed model, a proprietary dataset was constructed that could be fully adapted to the problem posed.

The Tweepy library (*Roesslein, 2020*) has been used to build this dataset. This library makes use of the Twitter API. The search patterns used for the construction of the dataset.

A keyword search was performed. These words are shown in Table 2, as well as their English translation:

A dataset of 26,143 tweets dating between 2nd November 2020 and 21st December 2020 was generated. Although for training purposes it is only necessary to have the text of the tweet and its classification, in order to carry out an analysis of the data, it was decided to store additional information such as Location, Number of followers, Number of followings, Total number of tweets, Date of user creation, Date of tweet creation, Number of retweets and Hashtags.

**Table 1 Summary of the objectives and weaknesses of the literature review conducted.**

| Reference | Findings | Weaknesses |
|---|---|---|
| (*Ahmad et al., 2018*; *Hasan, Maliha & Arifuzzaman, 2019*; *Lakshmi, Divya & Valarmathi, 2018*; *Br Ginting, Irawan & Setianingsih, 2019*) | Studies that corroborate good results in the task of text classification using conventional machine learning techniques (SVMs, Logistic regression). | These articles are not focused on obtaining predictive models for classifying categories of racism and xenophobia in Spanish texts. |
| (*Pereira-Kohatsu et al., 2019*) | HaterNet: dataset of 6,000 labelled tweets as hate speech or not. LDA, QDA, Random Forest, Ridge Logistic Regression, SVM and an LSTM combined with an MLP applied to HaterNet. | Tweets labelled as racist, but not racist. Very small percentage of tweets are related to racism, which is one of the topics that interests us for this work. |
| (*Basile et al., 2019*) | HateEval: two datasets, one in english and the second one in Spanish. Spanish dataset has 5,000 labelled tweets in the training set. | Only 1,971 tweets are related to racism and 987 of this subset were mislabelled. Tweets from some people dnouncing racism that are labelled as racist. |
| (*Nedjah, Santos & de Macedo Mourelle, 2019*; *Roy et al., 2020*) | Convolutional neural networks applied to different problems. | CNN applied to different problems, not only for text classification and not focused on racism and xenophobia. |
| (*Paetzold, Zampieri & Malmasi, 2019*) | Theory about recurrent neural networks (RNN) | Usually, training the BERT model from scratch on similar dataset could produce much better result (*Sany et al., 2022*; *Shahri et al., 2020*). |
| (*Talita & Wiguna, 2019*; *Bisht et al., 2020*; *Zhao et al., 2020*; *Zhang et al., 2021*) | Long Short-Term Memory Network (LSTM) applied to different problems. | Not all of them are focused on text classification. Those that are, do not make comparisons with BERT. |
| (*Plaza-Del-Arco et al., 2020*) | SVM, linear regression (LR), Naive Bayes (NB), decision trees (DT), LSTM and a lexicon-based classifier applied to a dataset composed by tweets related to xenophobia and misogyny. | The tweets are labelled with value 1 if speaks about xenophobia or misogyny. This may bias the results of the model. The best model was 74.2% of F1-score. |
| (*del-Arco et al., 2021*) | LSTM, Bidirectional Long Short-Term Memory Networks (Bi-LSTM) and CNN, mBert, XLM and BETO applied to HatEval and HaterNet. | The dataset are biased because the subsets were mislabelled. |

As far as data privacy is concerned, it is demonstrated that Cambridge Analytica is still alive and we can export people's behavioural characteristics without their consent just by acquiring publicly available data (*Pitropakis et al., 2020*; *Kandias et al., 2013*; *Isaak & Hanna, 2018*). This information, being public and anonymized, is exempt from the request for approval by an ethics committee (*Eysenbach & Till, 2001*).

## Preprocessing and labelling data

After analysing the different articles in which predictive models are generated, it was decided to make use of a subset of data consisting of 2,000 tweets to be labelled in the two categories to be classified: racist tweets and non-racist tweets. The subset of data chosen was balanced, with 52% of tweets labelled as non-racist and 48% of tweets labelled as racist.

From the initial set of 26,143 tweets, a pre-selection of 2,000 tweets was made by four students of computer engineering interpreting 50% of them as belonging to each category: racist or xenophobic tweets and tweets that were neither racist nor xenophobic. Subsequently, labelling was carried out manually by eight experts in the field of psychology. The set of 2,000 tweets was divided into four subsets of 500 tweets that

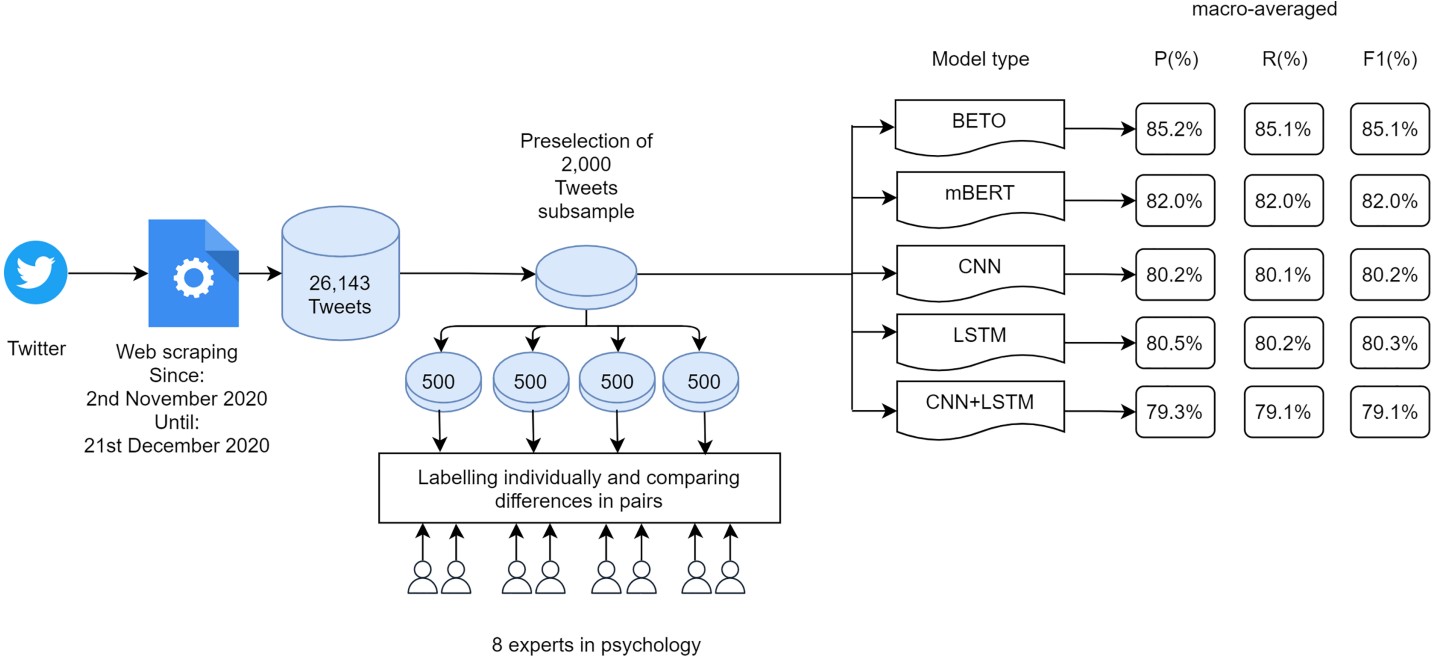

**Figure 1 Workflow of the research conducted.**

**Table 2 Spanish keywords used to collect dataset and their translation to English language.**

| Spanish keyword | Translation to English |
| --- | --- |
| moro, mora | moor |
| gitano, gitana, gypsy | gypsy |
| puto simio | fucking ape |
| negro de mierda, negra de mierda, puto negro, puta negra | fucking nigga |
| inmigracion, inmigrante | immigration, immigrant |
| patera | small boat |
| mena (menores extranjeros no acompanados) | unaccompanied foreign minor immigration, immigrant |
| panchito, panchita, sudaca | spic, greaser |

were labelled by two people individually. The pairs of experts then compared their labels and decided which label to assign to tweets where there might be some doubt as to whether or not they belonged to a category. Subsequently, a clustering phase of all tagged tweets was carried out.

Before generating this subset, a pre-processing of the data was carried out, which included the following tasks:

- Removal of extremely short tweets in which it is totally impossible to identify the presence or absence of racism.
- Removal of ironic tweets.
- Removal of tweets that are excessively badly written, making them difficult to understand or simply using invalid characters repeatedly.

- Conversion of all text to lowercase.
- Removal of URLs.
- Elimination of unnecessary spaces.
- Elimination of user names.
- Elimination of unnecessary characters.
- Elimination of accents.
- Elimination of stopwords.

The following criteria were also taken into account when selecting the subset:

- Variety of terms in the dataset. While the filters used to capture the tweets are extremely varied, in practice some terms are much more popular than others. For example, the terms "inmigracion" (immigration) or patera (small boat) appear in a much higher number of tweets than the term "negra de mierda" (fucking nigga). Therefore, if this factor is not taken into account when tagging tweets, we end up with a data set that is too poor with data that is too similar.
- Variety of meanings within the same term. Although many of the terms do not leave any doubt about their meaning, some of them may have different meanings which, if they are not both included, the behaviour of the model in the future when receiving input from these terms would be considerably reduced. An example of words to which this criterion applies directly is: "mora" (moor), "mena" (unaccompanied foreign minor) or "panchito" (spic).

## BERT models

BERT (*Devlin et al., 2019*) makes use of Transformers (*Vaswani et al., 2017*), a deep learning model proposed by researchers at Google and the University of Toronto in 2017 that has particular application in the field of natural language processing.

Like Recurrent Neural Networks (RNN), Transformers is designed to work with sequential data. In a similar way to humans, it is capable of processing natural language, serving to carry out tasks as diverse as translation or text classification. However, unlike RNNs, Transformers do not require sequential data, text in this case, to be processed in order.

This means that, when receiving a text as input, it is not necessary to process the beginning of the text before the end, which allows for much greater parallelisation and, therefore, reduces training times considerably.

Transformers have been designed using the concept of the attention mechanism, which itself was designed to memorise long sentences in machine translation tasks.

At the architectural level, it is based on an encoder-decoder architecture in which the encoders consist of a set of encoding layers that iteratively process the input layer by layer. In turn, the decoders consist of a set of decoding layers that do the same at the output of the encoder.

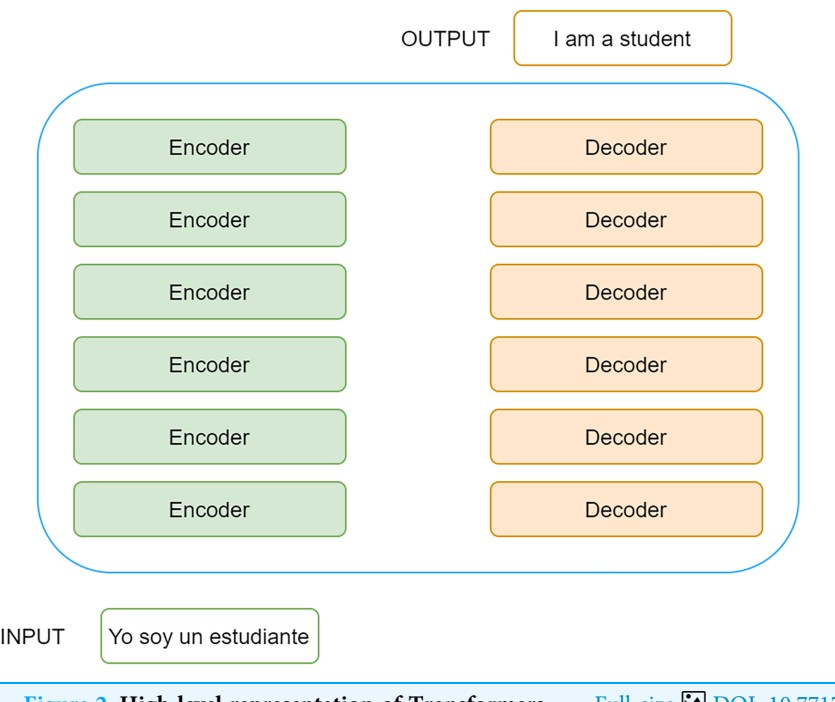

**Figure 2** **High-level representation of Transformers.**

Thus, when Transformers receive a text, it passes through a stack of encoders. The output obtained at the last encoder is passed to each of the decoders that make up the stack of decoders, resulting in a final output. A very high-level representation of Transformers can be found in the Fig. 2.

Each encoder consists of two main components, an attention mechanism called self-attention and a feed forward neural network.

The encoder receives a list of numeric vectors as input that flows through the self-attention layer. This layer helps the encoder to take into account the rest of the words in the sentence before encoding each word. Although the example in the figure above is very simple, the encoder, before encoding the word "Thinking", would take into account that this word is accompanied by the word "Machines" when generating the $z_1$ vector, just as it would do the same with the word "Machines" when generating the $z_2$ vector.

The output obtained for each of the input vectors, a list of z vectors in this case, is passed through a pre-fed neural network, which will generate an output for each z input vector. This output, which in the figure is represented as r, becomes the input to the next encoder, which will perform the same process as described above.

Finally, it should be noted that each encoder has a series of residual connections that prevents the output of one layer from being processed by the next layer. These mechanisms are particularly relevant in neural networks that have many hidden layers, allowing certain layers to be left unused if necessary. In addition, it should be noted that a normalisation process is applied to the output of each layer.

Since its publication, it has become the best existing solution for a large number of NLP problems. The most special feature of the model is the way in which it deals with the

meaning of words depending on the context in which they are used, using an architecture based on Transformers, which will be explained in due course.

BERT's first differentiating factor is the difference from older embedding generation techniques such as Word2Vec, Glove or FastText.

The embeddings generated using the aforementioned techniques are context-independent, *i.e.*, each word has a unique vector that defines it whatever the context in which it is used. This means that all the different meanings of a given word are combined into a single vector.

## Metrics to evaluate results

To evaluate the results obtained in each of the models developed, three different metrics have been used: Precision (P), Recall (R) and F1-score (F1).

### Precision

Precision is a metric used to calculate what percentage of the positive samples (which in this particular case equals the samples labelled as racist) have been properly classified. The formula used to calculate this metric is as follows, where c is equal to the class (0 = Non-racist, 1 = Racist), TP = True positive, FP = False positive and FN = False negative:

$$P(c) = \frac{TP}{TP + FP} \tag{1}$$

### Recall

Recall, in turn, is a metric used to calculate what percentage of the samples classified as positive have been properly classified. The formula used to calculate this metric is as follows, where c is equal to the class (0 = Non-racist, 1 = Racist), TP = True positive and FN = False negative:

$$R(c) = \frac{TP}{TP + FN} \tag{2}$$

### F1-score

F1-score is a metric used to calculate the effectiveness of a classifier by taking into account its accuracy and recall values. F1 assumes that the two metrics used are of equal importance in calculating effectiveness. If one of them is more important than the other, a different formula $F_\beta$ would have to be used. The formula used to calculate this metric is as follows, where P equals the precision value and R equals the recall value:

$$F_1 = \frac{2 * P * R}{P + R} \tag{3}$$

## EXPERIMENTS AND RESULTS

### Experimental setup

As part of the development of this project, a total of five different predictive models have been developed. Two are transfer learning models and three are deep learning models.

Within the transfer learning models, a model based on BETO has been developed, which is very similar to BERT trained in Spanish, and a model based on mBERT, a BERT checkpoint that can be used on texts in a large number of different languages. With respect to deep learning models, a model has been developed that implements a convolutional neural network, another model that implements a recurrent neural network (LSTM) and a last model that fuses a convolutional neural network with another recurrent neural network.

### BETO model

BETO (*Cañete et al., 2020*) is a transfer learning model that has been trained in the same way as BERT, except that it has been trained using texts written in Spanish instead of English, including Wikipedia entries, subtitles of series and movies, and even news. The specific technique with which it has been trained is called Whole Word Masking, a technique very similar to the Masked Language Model in which instead of hiding random tokens, it ensures that the hidden tokens always constitute a word, which means that if a token corresponding to a sub-word is hidden, the rest of the tokens that make up the whole word are also hidden. If we make a comparison between BETO and the different BERT models, we could say that it is a model extremely similar to the base BERT, given that it has 12 layers of encoders in its architecture.

Within BETO, there are two different models, one that has been trained using words containing both upper and lower case letters (original texts therefore) and another one that has been trained using only words written in lower case letters (texts are processed to comply with this premise). Although both models offer great results, depending on the problem to be solved, it is more appropriate to use one rather than the other. In this particular case, given that the dataset used is made up of tweets (where people are not at all careful about the way they express themselves), the uncased model is the one that performs better. After carrying out a large number of tests with both models, it has been found that the results obtained by this model are between 0.5% and 1% better, which justifies the selection of this model.

### mBERT model

mBERT (Multilingual BERT) (*Devlin et al., 2019*) is a transfer learning model that has been trained on Wikipedia texts written in 104 different languages, including Spanish. Like BETO, it also has an identical architecture to the BERT base, with 12 layers of encoders, and two different models, one trained with texts containing both lowercase and uppercase letters and the other trained with texts containing only words written in lowercase letters. Contrary to BETO, after several tests, it has been found that the cased model performs slightly better than the uncased model. However, due to some problems in the

tokenisation process, it has been necessary to convert all text to lower case even though the selected model is case-sensitive.

Multilingual models frequently encounter the common problem of language detection. This type of model does not usually have any mechanism or system that allows the detection of the language in which the texts that make up the input are written, which means that on many occasions the tokenizer makes mistakes when dividing the texts into tokens. If we add to this the fact that this type of model does not have mechanisms that allow words from different languages with the same meaning to be represented in a similar way in the vector space, everything seems to indicate that the results obtained after training these models are worse than those obtained by native models, that is, those designed to work with texts written in a single language (as is the case of BETO in Spanish or BERT in English).

### CNN model

This model is made up of the following layers:

- Embedding layer.
- Three convolutional layers each followed by a MaxPooling layer. A total of 256, 128 and 64 filters. MaxPooling of $2 \times 2$.
- Final output layer with a single neuron, in charge of classifying the sample.

The schematic architecture of the model is shown in Fig. 3.

### LSTM model

This model consists of the following layers:

- Embedding layer.
- LSTM layer with 64 units.
- Two fully connected layers with dropout between them.
- Final output layer with a single neuron, in charge of classifying the sample.

The schematic architecture of the model is shown in Fig. 4.

### CNN + LSTM model

This model consists of the following layers:

- Embedding layer.
- Convolutional layer with 64 filters.
- LSTM layer with 128 filters.
- Final output layer with a single neuron, in charge of classifying the sample.

The schematic architecture of the model is shown in Fig. 5.

## Transfer learning parameter optimisation (BETO and mBERT)

In order to ensure that the results obtained by the transfer learning models are as high as possible, a series of tests have been carried out in which the performance of

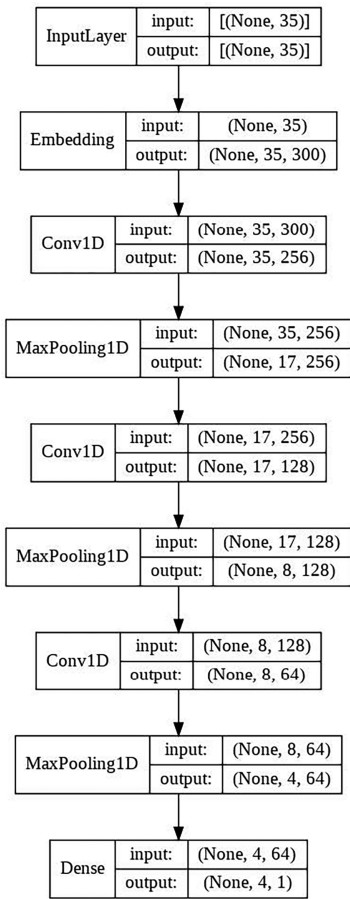

**Figure 3 CNN architecture scheme.**

both models has been tested as a function of the value taken by the various hyperparameters.

Although the number of hyperparameters that can be customised is really high, in practice it is totally impossible to try to modify all of them given the exponential growth experienced in the number of combinations. Therefore, it has been necessary to select those parameters that have a greater relevance in the behaviour of the model (*Sun et al., 2019*).

The parameters selected were as follows:

- Type of model: Cased or uncased.
- Number of epochs: 2, 4 and 8. The number of epochs dictates the number of times the model will process the entire training set.
- Batch size: 8, 16, 32 and 64. This parameter indicates the number of samples to be processed by the model until an internal update of the model weights is performed.
- Optimiser: Adam and Adafactor. Mechanisms used to manage the update of the model weights.

- Learning rate: 0.00002, 0.00003 and 0.00004. Parameter that determines the step size when performing an update in the model with a view to approaching a minimum in a loss function.

As the dataset is made up of tweets, the length of which is generally limited to between three and 35 words, the maximum sequence size is not so relevant. However, if we were working with other types of text, it would be another parameter to be modified.

The different possible combinations tested (a total of 144) and the best parameters found for each of the two models can be found in the Table 3.

The most optimal configurations for both models are practically identical, with the exception of the model type and number of epochs. While in the case of BETO it is the uncased model that obtains the best results, in mBERT it is the cased model that obtains the best results, although the difference is minimal. In turn, in relation to the number of epochs, BETO obtains better results with 8 and mBERT with 4.

## Optimisation of deep learning parameters (CNN, LSTM and CNN + LSTM)

In order to ensure that the results obtained by the deep learning models are as high as possible, a series of tests have been carried out in which the performance of the models has been tested depending on the value of certain strategically selected parameters.

As in BERT, the number of parameters to be modified is really high, much higher still in this case since some factors such as the number of hidden neurons in each layer greatly influence the performance of the models. Although several tests have been carried out on both the structure of the different models and the number of hidden neurons in each layer, it is more important to focus on the remaining parameters that have been modified.

The selected parameters have been the following:

- Batch size: 16, 32, 64, 128.
- Dropout: 0.25, 0.5.
- Optimizador: Adam, SGD.
- Funcion de activacion: Relu, Tanh.
- Tasa de aprendizaje: 0.01, 0.02, 0.001, 0.002.

On numerous occasions, the number of epochs is also a parameter to be modified. However, in all the models pertaining to this project, we have chosen to use a mechanism called EarlyStopping, which is a regularisation mechanism used to avoid overfitting and which consists of stopping training when the control metrics of the validation set begin to decay.

The different possible combinations tested (a total of 64 in the CNN model and 128 in the LSTM and CNN + LSTM models) and the best parameters for each of the three models can be found in Table 4.

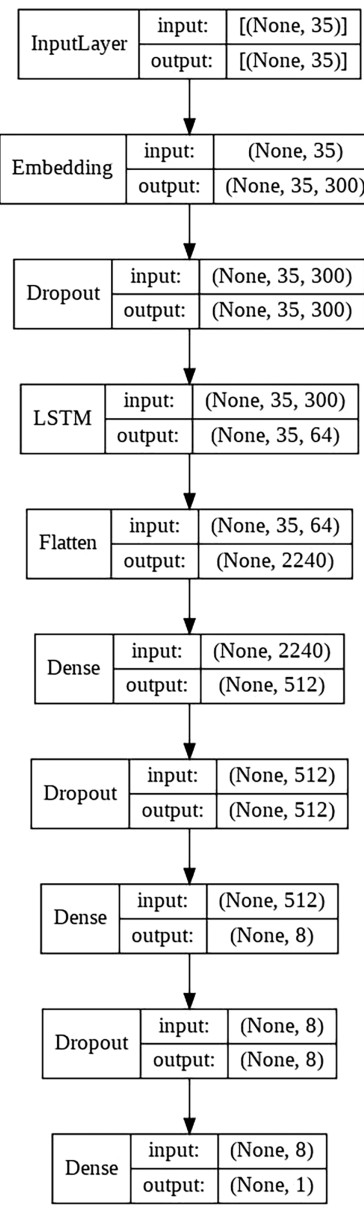

**Figure 4 LSTM architecture scheme.**

The most optimal configurations for the three models are very similar, sharing values in optimiser, activation function and learning rate. Dropout, which is not present in the proposed CNN architecture, is identical in the other two models.

## Hardware and Software used for the experiments

To perform the pre-processing of the tweets and apply the deep learning techniques, a Jupyter notebook, Python 3.6 was used and run on a computer with the following characteristics: Intel(R) Core(TM) i7-9700K CPU @ 3.60GHZ, 32.0GB RAM and an NVIDIA GeForce RTX 2080 6GB graphics card. The spaCy and nltk libraries were used to pre-process the text content.

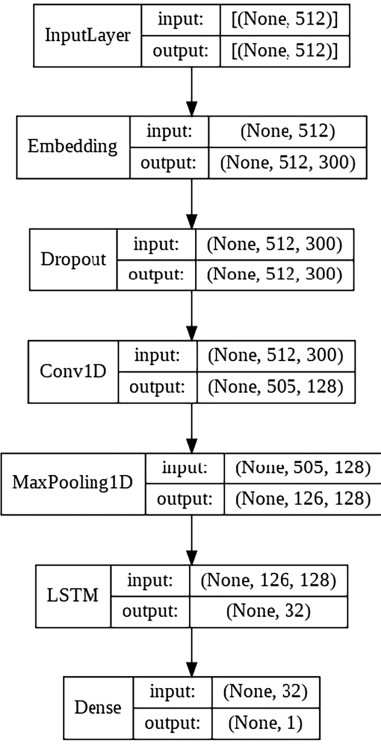

**Figure 5 CNN + LSTM architecture scheme.**     

**Table 3 The best transfer learning hyperparameters.**

| Hyperparameter | Options | BETO | mBERT |
|---|---|---|---|
| Model type | (cased, uncased) | uncased | cased |
| Epochs | (2, 4, 8) | 8 | 4 |
| Batch size | (8, 16, 32, 64) | 8 | 8 |
| Optimizer | (Adam, Adafactor) | Adam | Adam |
| Learning rate | (2e−5, 3e−5, 4e−5) | 4e−5 | 4e−5 |

**Table 4 The best deep learning hyperparameters.**

| Hyperparameter | Options | CNN | LSTM | CNN + LSTM |
|---|---|---|---|---|
| Batch size | (16, 32, 64, 128) | 32 | 64 | 64 |
| Dropout | (0.25, 0.5) | N/A | 0.5 | 0.5 |
| Optimizer | (Adam, SGD) | Adam | Adam | Adam |
| Activation function | (Relu, Tanh) | Relu | Relu | Relu |
| Learning rate | (1e−2, 2e−2, 1e−3, 2e−3) | 2e−3 | 2e−3 | 2e−3 |

## Results

Table 5 shows the results obtained after numerous runs of the various predictive models. The runtime column refers to the time spent training and validating the model.

**Table 5  Results obtained for all models.**

| Model | Non-racist | | | Racist | | | Macro-averaged | | | Runtime |
|---|---|---|---|---|---|---|---|---|---|---|
| | P (%) | R (%) | F1 (%) | P (%) | R (%) | F1 (%) | P (%) | R (%) | F1 (%) | |
| BETO | **84.28** | **87.30** | **85.76** | **86.17** | 82.94 | **84.52** | **85.22** | **85.12** | **85.14** | 1,230 s |
| mBERT | 83.28 | 81.11 | 82.18 | 80.73 | **82.94** | 81.82 | 82.00 | 82.02 | 82.00 | 1,129 s |
| CNN | 80.13 | 81.43 | 80.78 | 80.21 | 78.84 | 79.52 | 80.17 | 80.14 | 80.15 | 840 s |
| LSTM | 78.90 | 84.04 | 81.39 | 82.05 | 76.45 | 79.15 | 80.48 | 80.24 | 80.27 | 844 s |
| CNN + LSTM | 77.58 | 83.39 | 80.38 | 81.11 | 74.74 | 77.80 | 79.34 | 79.07 | 79.09 | 938 s |

**Note:**
The best results are highlighted in bold.

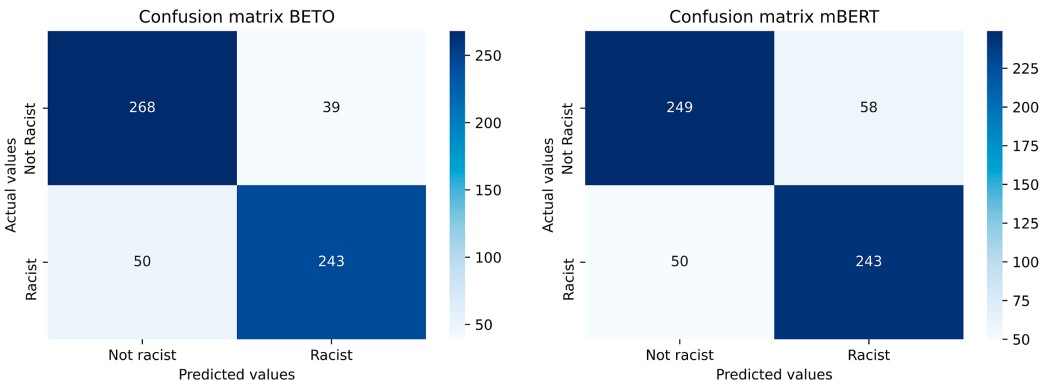

**Figure 6  Confusion matrices of the best performing models (BETO and mBERT).**

BETO offers the best results in each and every one of the metrics calculated, obtaining a wide advantage over the rest of the models. The confusion matrices of the two best-performing models are shown in Fig. 6.

As regards to the most important metric of all, Macro F1-score, in which all classes have the same importance in determining the effectiveness of the model, BETO obtains 85.14%, improving by more than three points over the other transfer learning model developed (mBERT) and between five and six points over the deep learning models. Several conclusions can be drawn from these results.

# CRITICAL EVALUATION AND COMPARISON OF THE RESULTS

Firstly, comparing the two transfer learning models, it can be seen that the native model (BETO) performs substantially better than the multilingual model (mBERT). The main reasons for this difference between the two models are as follows:

- Vocabulary difference between the two models: As BETO was trained with texts written in Spanish and mBERT with texts written in 104 different languages, the number of Spanish words in the two models differs greatly, which means that the percentage of words in the dataset present in the models' vocabulary of the models is also very different. As these are transfer learning models, where a generic model is used to solve a

specific problem, the difference in the word coverage of the dataset has a significant impact on the models' results.

- Difference in the tokenisers: While BETO has a tokeniser that is solely and exclusively responsible for tokenising texts written in Spanish, mBERT has a generic tokeniser that does not even have a mechanism for detecting the language in which the text it is processing is written. This means that on numerous occasions tokenisation in mBERT is carried out erroneously by recognising Spanish words as words that exist in other languages.

With respect to the deep learning models, although they perform considerably worse than the transfer learning models (especially BETO), it should be noted that the results obtained are really good if we take into account the simplicity of the models developed and the small amount of data in the training set (only 1,400 samples).

Among the factors that influence the poorer performance of these models are the following:

- Single embeddings for several meanings of the same word: Although pre-trained embeddings are used which have a coverage of 82.4% of the words (an exceptionally good figure), all meanings of the same word are represented by the same numeric vector. This means that, for words in this situation, their numeric vector is influenced by all the different meanings of the word, which causes the accuracy of the model to be reduced slightly.
- Context independence: Unlike transfer learning models, deep learning models do not have such an effective mechanism for representing words based on their context (RNNs have one, but it is nowhere near as effective as that implemented by BERT-based systems). This makes it often a bit complex for such models to take into account a word into account its specific context, which in addition to the unique embeddings for each word leads to poorer model results.

Taking into account all of the above into account, it is worth highlighting that BETO is, without a doubt, the model that best solves the problem posed in this research, also demonstrating how important it is to develop native transfer learning models, as they obtain better results than multilingual models.

In our research, BETO was the best performing model. Deep learning models generally operate as black boxes, so in some cases it is difficult to reason why some perform better or worse. However, our performance results are consistent with those obtained in other studies (*del-Arco et al., 2021*; *Gutiérrez-Fandiño et al., 2021*; *Pomares-Quimbaya, López-Úbeda & Schulz, 2021*) in which BETO was also the best performing model. Although BETO and mBERT have very similar architectures, BETO was trained on Spanish data and mBERT was pre-trained on 104 languages. In this case it is evident, as in other articles that also make use of them (*Gutiérrez-Fandiño et al., 2021*; *Pomares-Quimbaya, López-Úbeda & Schulz, 2021*) that for the present problem, the model trained specifically with the same language as the dataset for which an efficient solution is to be found offers better results.

## CONCLUSIONS

In this research, the two objectives initially set have been completed:

- Messages on the Twitter platform containing words related to racism were obtained. A subsample of 2,000 messages was tagged, resulting in a balanced dataset.
- Different predictive models were generated using NLP techniques. These models were based on deep learning (CNN, LSTM and CNN + LSTM) and on transfer learning models (BERT). The best performing model was based on BERT, namely BETO with a precision of 85.22%.

This fact justifies the need for the development of native transfer learning models. Having been trained with texts written in a single language instead of dozens of languages, the vocabulary of native models is far superior to that of multilingual models, which translates into greater effectiveness in the vast majority of situations.

This research shows preliminary results that need to be further investigated by improving some weaknesses, *e.g.*, the size of the dataset used. Future research will increase the dataset, to achieve a more robust validation of the model presented in this article. Following more robust validation, it is intended to add new languages and to integrate these models into web applications that can be useful to society. Other limitations include, as in other studies that present text classification models, the need to retrain the model to include new terms generated by society over time. In addition to this, it is proposed to implement the model in an application that helps to automatically detect racism in different websites and to present results and validation of the complete system.

### Funding

This research was funded by the Junta de Castilla y León grant number LE014G18. The funders had no role in study design, data collection and analysis, decision to publish, or preparation of the manuscript.

### Grant Disclosures

The following grant information was disclosed by the authors:
Junta de Castilla y León: LE014G18.

### Competing Interests

The authors declare that they have no competing interests.

### Author Contributions

- José Alberto Benítez-Andrades conceived and designed the experiments, analyzed the data, performed the computation work, authored or reviewed drafts of the paper, and approved the final draft.

- Álvaro González-Jiménez conceived and designed the experiments, performed the experiments, performed the computation work, prepared figures and/or tables, and approved the final draft.
- Álvaro López-Brea conceived and designed the experiments, performed the experiments, performed the computation work, prepared figures and/or tables, and approved the final draft.
- Jose Aveleira-Mata performed the experiments, performed the computation work, authored or reviewed drafts of the paper, and approved the final draft.
- José-Manuel Alija-Pérez performed the experiments, performed the computation work, authored or reviewed drafts of the paper, and approved the final draft.
- María Teresa García-Ordás conceived and designed the experiments, analyzed the data, performed the computation work, prepared figures and/or tables, authored or reviewed drafts of the paper, and approved the final draft.

## Data Availability

All the data used in the experiments, the models generated and the source code are available at Zenodo: Álvaro González Jiménez, & Jose Alberto. (2022). Detecting racism and xenophobia using deep learning models on Twitter data: CNN, LSTM and BERT Latest (Version V2). Zenodo. https://doi.org/10.5281/zenodo.5886624.

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
