# Peer review of "Detecting racism and xenophobia using deep learning models on Twitter data: CNN, LSTM and BERT"

_PeerJ Computer Science, doi:10.7717/peerj-cs.906_

## Round 0.1 · original submission · Major Revisions

This is an interesting manuscript, so please consider the reviewers' comments in order to improve the quality.

Reviewer 1 ·

Basic reporting

The authors of this paper have focused on a very important aspect of our society, the increasing popularity of racism and xenophobia over social media. More specifically, the authors focus on the Twitter platform and the Spanish language where they use three different deep learning models, CNN, LSTM, and BERT with the last one being the most effective during experimentation.
The paper is well structured and well written.

This is another important paper in the area that utilizes NLP and behavioural characteristics, showcasing existing problems. However, there are a few issues that the reviewer would like to raise with the authors.

The contributions of the paper are three, the assembly of the datasets, the experimentations on top of the datasets, and last the critical comparison and analysis.
A table at the end of the related literature can help the authors summarize their novelties and the potential reader understand them.

Experimental design

The hardware and software specifications of the testbed environment are not mentioned and they should be for the replicability of the results.

Validity of the findings

The discussion section should be renamed to critical evaluation and comparison of the results. In the same section, the authors should further explain their results, not just demonstrate the numbers. They should clarify why each methodology had different results and how they can be used potentially in an automated environment.

There is no mention of introduced overhead if any between the different deep learning models.

Additional comments

There is no mention of privacy. This work clearly demonstrates that Cambridge Analytica is still alive and we can export people’s behavioural characteristics without their consent just by acquiring publicly available data (e.g. https://doi.org/10.1109/MC.2018.3191268 , https://doi.org/10.1109/UIC-ATC.2013.12 , https://doi.org/10.3390/make2030011 ).

·

Basic reporting

This manuscript proposes several deep learning models to classify Spanish texts on xenophobia and racism. Among the objectives of the research presented are to generate a dataset of Spanish-language tweets labelled as xenophobic or racist, or not. To carry out this research, the authors have compiled their own dataset which they have shared with the rest of the scientific community and, on this, they have applied CNN, LSTM and BERT models. The manuscript is well-structured, state-of-the-art, scientifically justified, reproducible and novel.
The research focuses on generating models with a Spanish dataset which, a priori, is often difficult to justify. However, the authors mention other similar research such as HaterNet and HatEval, highlighting the possible weaknesses of these experiments and thus justifying the experiments presented in this manuscript.

Experimental design

At the methodological level, the authors mention the methodology applied, how they obtained the dataset and, in addition, they have shared the jupyter notebooks containing the code of the models generated and the results obtained.

Validity of the findings

The authors need to address some minor weaknesses before the manuscript can be considered for publication in a journal.

1. Although CNN, LSTM and BERT techniques are indicated in the title, CNN and LSTM techniques are not mentioned in the abstract. It is recommended that the authors modify the abstract by adding this information and other interesting information regarding the final results obtained, for example, the % accuracy obtained in the models.

2. To improve the interpretability of the results, authors are advised to add some of the figures of the confusion matrices corresponding to the models that obtained the best results.

3. On the other hand, they should revise the manuscript and modify some concepts that appear written in two different ways, for example "tagged" and "labelled".

Reviewer 3 ·

Basic reporting

This paper evaluates different approaches to detect hate speech motivated by racism or xenophobia in Spanish using Twitter data. The structure of the manuscript is clear and there is an extensive literature review. The authors justifies the need of an adapted model for supervised text classification in Spanish to detect hateful messages particularly aimed at migrants or non-caucasian people.

I consider however that the rational is still poor and needs to create arguments in the next directions:
-If the problem is language-based, what other developments have shown that native models for hate speech detection are better that general ones? This is, showing examples of attempts in languages different from English that have a good performance (and not only in Spanish).
-Compare the results of this paper to the best performances obtained in other languages.

This would make the manuscript more useful for international audiences.

Regarding previous attempts, the authors mention HaterNet and HatEval, but are missing Pharm. This project develops deep learning models to classify tweets with hate speech towards migrants and refugees in Spanish, Greek and Italian. In Spanish, they manually labelled more than 12,000 tweets finding 1,390 hateful messages to build the classification model with RNN. This approach does not use BERT, but still gets good metrics (f1-score=0.87 for Spanish). The classification interface (http://pharm-interface.usal.es) offers more information (probably the training corpus under request too) and these two papers:
Vrysis, L.; Vryzas, N.; Kotsakis, R.; Saridou, T.; Matsiola, M.; Veglis, A.; Arcila, C.; Dimoulas, C. (2021). A Web Interface for Analyzing Hate Speech. Future Internet, 13(3), 80. https://doi.org/10.3390/fi13030080
Arcila, C., Sánchez, P., Quintana, C., Amores, J. & Blanco, D. (2022, Online preprint). Hate speech and social acceptance of migrants in Europe: Analysis of tweets with geolocation. Comunicar, 71.

On the other hand, what is the goal of point 2.2? Why is it called "Sentiment analysis"? Do the authors refer probably to text classification? The title is confusing since SA is not the scope of this paper.

Experimental design

The method is correctly included in the paper and shows that the research is in line with the scope of the Journal. The models are well described, although I am not sure if most of the details are necessary for a Computer Science specialized audience (i.e. description of what BERT is or definition of te evaluation metrics).
The training corpus seem to be labelled by specialists in Psychology and has specific examples of racism and xenophobia. This is a good value of the paper. Though, the reliability of this manual classification is not reported (i.e. inter-coder reliability measure), which seriously compromise the quality of the ad hoc data set. This is extremely important since this test can tell if different coders were really considering the same as hate, and then validated the qualitative category.
In addition, there is a relevant privacy concern in the Twitter data. In Zenodo, the tweets include the usernames and the given label (hate/ no hate). This does not meet the ethical and data protection standards for Twitter analysis.

Validity of the findings

The findings are relevant for the hate speech detection field, since they can be used to build better models in different languages. In special, the use of BETO and the verification of its enormous advantage can help other researchers and practitioners to create better models in the future.
My concern is the validation of the results in other different data. This is, for example, collecting new Twitter data (within another timeframe or word filters) and validate the obtained models. I consider this external validation phase extremely important for ML models.
Regarding the conclusions, I think that the limitations and future research should be better developed. Is just the size of the dataset the only limitation? What specific applications can generate this model? How can the model deteriorate in time with the use on new words/sentences? How often this model should be re-trainned to be really useful in real-life applications?

Additional comments

None

---

## Round 0.2 · accepted · Accept

I read Reviewer 3 reference and I consider that does not worth it to be included in your manuscript. Please, do not include this reference in your manuscript.

Finally, congrats on the acceptance of your manuscript!

Reviewer 1 ·

Basic reporting

The paper was already clear and its quality was improved according to the comments.

Experimental design

The experimental design was improved according to the comments. New measurements were added as requested.

Validity of the findings

The impact of the findings was addressed and extra discussion was added.

Additional comments

All comments were addressed.

·

Basic reporting

The authors have accomplished all the suggested improvements and changes.
Under my opinion the paper must be accepted in the current state.

Experimental design

ok

Validity of the findings

ok

Additional comments

None

Reviewer 3 ·

Basic reporting

The authors have addressed my main concerns and the manuscript is much better now. The missing reference can be found in:
Arcila, C., S nchez, P., Quintana, C., Amores, J. & Blanco, D. (2022, Online preprint). Hate speech and social acceptance of migrants in Europe: Analysis of tweets with geolocation. Comunicar, 71. https://www.revistacomunicar.com/index.php?contenido=detalles&numero=71&articulo=71-2022-02&idioma=en

Experimental design

The authors have addressed my main concerns in the experimental design.

Validity of the findings

The authors have addressed my main concerns in the validity of findings.

Additional comments

None